# Mechanisms of Spontaneous Curvature Inversion in Compressed Graphene Ripples for Energy Harvesting Applications via Molecular Dynamics Simulations

**DOI:** 10.3390/membranes11070516

**Published:** 2021-07-09

**Authors:** James M. Mangum, Ferdinand Harerimana, Millicent N. Gikunda, Paul M. Thibado

**Affiliations:** Department of Physics, University of Arkansas, Fayetteville, AR 72701, USA; jmmangum@uark.edu (J.M.M.); fharerim@uark.edu (F.H.); mngikund@uark.edu (M.N.G.)

**Keywords:** graphene, ripples, thermal motion, energy barrier, strain, energy harvesting, molecular dynamics

## Abstract

Electrically conductive, highly flexible graphene membranes hold great promise for harvesting energy from ambient vibrations. For this study, we built numerous three-dimensional graphene ripples, with each featuring a different amount of compression, and performed molecular dynamics simulations at elevated temperatures. These ripples have a convex cosine shape, then spontaneously invert their curvature to concave. The average time between inversion events increases with compression. We use this to determine how the energy barrier height depends on strain. A typical convex-to-concave curvature inversion process begins when the ripple’s maximum shifts sideways from the normal central position toward the fixed outer edge. The ripple’s maximum does not simply move downward toward its concave position. When the ripple’s maximum moves toward the outer edge, the opposite side of the ripple is pulled inward and downward, and it passes through the fixed outer edge first. The ripple’s maximum then quickly flips to the opposite side via snap-through buckling. This trajectory, along with local bond flexing, significantly lowers the energy barrier for inversion. The large-scale coherent movement of ripple atoms during curvature inversion is unique to two-dimensional materials. We demonstrate how this motion can induce an electrical current in a nearby circuit.

## 1. Introduction

Recent developments in low power-consuming circuit designs have reduced input power to nanowatts in active mode and picowatts in standby mode. In addition, some applications operate with an ultra-low duty cycle, which further lowers the overall power requirement [1]. These breakthroughs make it possible to use ambient vibrations as a power source instead of batteries [2,3,4]. Consequently, there is growing interest in developing power sources that scavenge energy from the local environment [5,6,7,8].

Microelectromechanical systems (MEMS) are an attractive approach to energy harvesting. For example, a 14 nm thick silicon nitride cantilever is extremely flexible and will vibrate under the slightest influence. The kinetic energy of the cantilever can be converted into stored electrical charge using an electrical circuit with a rectifier and a capacitor. Materials thinner than silicon nitride would be better, as the flexural rigidity decreases with thickness to the third power. The ultimate thinness is found with two-dimensional (2D) materials and graphene is the most durable of these. A graphene membrane can be ten times thinner than silicon nitride and, therefore, 1000 times more flexible [9,10]. In addition, graphene has an extremely high electrical conductivity, thermal conductivity, and is non-magnetic. We exploit these properties by incorporating freestanding graphene into an electrostatic circuit such that the graphene membrane serves as the mobile plate of a spontaneously varying capacitor. As the graphene moves up and down, a displacement current is produced in a nearby electrical circuit with diodes that are used to charge storage capacitors. A schematic of our graphene energy harvesting circuit is shown in Figure 1. As the graphene membrane vibrates up and down, the distance to the fixed upper plate changes in time. This results in a time-varying capacitance. For a fixed bias voltage, V, charge will flow on and off the graphene capacitor as its capacitance changes in time [11]. When charge flows clockwise, it passes through diode one, D1, and charges storage capacitor one, C1. When charge flows counterclockwise, it flows opposite the bias voltage, recharging it, charges storage capacitor two, C2, then passes through diode two, D2. Even though the current is low and, therefore, the diode resistance is high, this circuit was found to have 50% efficiency when operated at its maximum output power [12]. In a recent study, the graphene circuit shown in Figure 1 was experimentally tested using suspended graphene near a fixed metal electrode and shown to induce an electrical current in the circuit. This result is the motivation for this work [13].

When graphene is suspended, the membrane can be held under tension or compression. When taut, the vibrational modes produced are the well-studied drumhead vibrations. For graphene, these modes have very high frequencies [14]. One potential application for taut membranes is a high Q sensor, an active area of research [15]. Alternatively, when graphene is compressed, a bump or ripple forms. Its movement is then governed by barrier crossing events and the vibrations shift to extremely low frequencies [16,17,18,19]. This area of research is in the early stages of development and is gaining attention for its potential applications [20,21,22,23].

When graphene was first suspended in the early 2000s [24,25,26,27,28], it was found to naturally form ripples due to its self-compression stability property [16,29,30,31,32,33,34,35,36]. The geometry of these ripples has been well-characterized in terms of average height and diameter. For example, using an in-plane magnetic field, it was found that ripples have a height of about 0.6 nm and a diameter at around 8 nm [37]. Using atomic force microscopy (AFM) on ripples held at 400 C, the height was measured at 0.13 nm with a diameter of about 20 nm [37]. Room temperature AFM studies found the height to be around 0.2 nm with a diameter of 64 nm [38]. Scanning tunneling microscope (STM) studies found the average height to be around 0.32 nm, with a diameter of about 15 nm [39].

Some dynamic properties of ripples have been studied using scanning probe microscopy (SPM). However, the shape transformation details of a ripple during curvature inversion cannot be imaged using SPM, as the process takes place over too large an area and too short a time for SPM. Molecular dynamics simulations (MDS), on the other hand, are an ideal method for investigating the evolution of ripple inversion. In this study, we used MDS to quantify the dynamics of graphene ripples as they invert their curvature. To achieve this, we first determined the energy required to compress and stretch graphene. Second, we used the strain–energy relationship to build a double-well potential for a particular ripple, then investigated the average inversion rate as a function of strain. This allowed us to report the energy barrier as a function of compressive strain. Finally, we derived a detailed picture of the shape transformations the ripple undergoes during the curvature inversion process through cross-sectional analysis.

## 2. Materials and Methods

We first created a flat circular lattice of about 7000 carbon atoms arranged in a honeycomb structure. The potential energy of the lattice was found for a number of different bond lengths in LAMMPS using the AIREBO C-C potential at zero Kelvin. Atomic pairs were set to interact within three standard deviations of their equilibrium distance [40,41]. The minimum energy of the lattice occurs at a bond length of 0.140 nm. For a lattice compressed or stretched by less than one percent, the potential energy in units of eV follows $\Delta U=\left(47\right)\;\epsilon^{2}$, where ϵ is the percent strain. 

We constructed three-dimensional (3D) ripples starting from the flat circular lattice and using a bond length smaller than 0.140 nm. Next, inside a radius of 7.5 nm, a third dimension was added to the carbon atoms using the following cosine function:(1)$$z=h\;cos\left[\frac{\pi }{4}\frac{1}{7.5}\sqrt{x^{2}+y^{2}}\right]$$
where h is the central height of the ripple and x and y are the coordinates of the carbon atoms. As h increases, the arc length in a vertical plane also increases, which stretches the bond lengths back to the original 0.140 nm and aids in the simulation’s optimization. We found that for an original planar strain of −0.7%, the relaxed height of the ripple is about 0.8 nm. An illustration of a ripple with this strain is shown in Figure 2a. An approximate formula relating the original planar percent strain to the relaxed final height in nm is given by:(2)$$\epsilon =-1.08\;h^{2}$$

This formula comes from a simple geometric analysis and computes the strain using the diameter and smooth arc length of the cosine ripple. After forming the ripple, we retain a 1.5 nm wide graphene annulus around the ripple at zero height. The annulus is always fixed in place, its position defines the x−y plane, and it can be seen in the 3D rendering shown in Figure 2a.

The zero Kelvin potential energy of a −0.7% strained ripple is the double-well potential, as shown in Figure 2b. The two minimums correspond to a convex- and concave-shaped ripple with equilibrium bond lengths, as also illustrated in Figure 2b. When the ripple is compressed into a flat lattice, its energy is 23 eV higher. This is the barrier height between the two minimums at zero height. If the ripple is stretched by 0.7%, starting with its relaxed cosine shape [42], its potential energy will increase by 23 eV and equal the barrier height. These five points precisely fit the double-well potential energy function given by:(3)$$U\left(z\right)=\Delta U\left[\;\frac{z^{4}}{z_{0}^{4}}-2\frac{z^{2}}{z_{0}^{2}}\;\right]$$
where ΔU is the barrier height, z is the central height of the cosine-shaped ripple, and $z_{0}$ is the central height of the ripple at its minimum energy. This potential energy function was also found using density functional theory (DFT) [5]. Note that the height of the energy barrier is about 1000 times larger than room temperature energy ($k_{B}T)$, where $k_{B}$ is Boltzmann’s constant and T is the absolute temperature. As a result, the ripple will have an extremely low probability of spontaneously inverting its curvature. However, these ripples do invert their curvature at room temperature, so it is likely that the shape is never flat as it goes through the process of inverting its curvature [43].

Next, we constructed numerous 15 nm diameter ripples, each featuring a different strain, using the method described above. After ripple construction, the ripple temperature was set to 3000 K and maintained at that temperature using a Nosé–Hoover thermostat. The equilibrium bond length changed from 0.140 nm at zero K to 0.143 nm at 3000 K. The equations of motion were integrated using a time step of 1 fs. The system was equilibrated for 0.5 ns starting from the initial cosine shape. The subsequent trajectory from a production run of 40 ns was used for analysis in this study.

## 3. Results

### 3.1. A. Bistable Ripples

The cross-sectional profile of a typical equilibrated cosine-like shaped ripple is shown in Figure 3a. This line profile features several kinks. When graphene is held at high temperatures, kinks appear because of its extremely flexible carbon bonds [44]. For this reason, the more meaningful strain values reported in this manuscript are found using Equation (2), which assumes a smooth arc.

We tracked the central height of the ripple in time. One such example is shown in Figure 3b. For this ripple, the strain is −0.78%. It is trapped in a concave shape for the entire 4 ns shown. Another ripple with strain −0.13% is shown in Figure 3c. In this case, the strain is too low to be bistable at the temperature of the simulation. When the ripple strain is between these extremes, it can invert its curvature multiple times within a 4 ns time interval. One example is shown in Figure 3d. In order to test the integrity of our simulation methods, we successfully reproduced similar ripple curvature inversion results previously published [43].

### 3.2. B. Average Time between Inversion

We narrowed the strain values to those that yield bistable ripples that invert their curvature on a time scale accessible for this study. We then created 10 more ripples with similar but varying strains and ran 40 ns MDS for each. For all these simulations, we found the average time between curvature inversions, then plotted this time versus the calculated strain found using Equation (2), as shown in Figure 4a. As expected, the larger the compressive strain, the longer the ripple stays on one side. The average time, 〈t〉, to cross a barrier height, ΔU, is given by Kramers’ rate [45]:(4)$$\langle t\rangle \propto e^{\frac{\Delta U}{k_{B}T}}$$

The best fit to the average time between flips yields $\Delta U=\left(3.4\right)\;\epsilon^{2}$ for the barrier height, which is shown in Figure 4b.

Using the energy barrier results shown in Figure 4b, we constructed the double well potential for a ripple with strain −0.7%. This ripple has an energy barrier of 1.6 eV, as illustrated in Figure 5a. DFT calculations for graphene membranes were carried out in another study and show a similar energy barrier to what we found [5]. This barrier height is 14 times smaller than the energy barrier for flat graphene featuring the same strain (23 eV). This leads to the conclusion that uniform flattening is not the mechanism behind inversion. The ripple could alter its shape to look like the illustration in Figure 5b. Here, the ripple maximum has been shifted to the right, as the other half of the ripple starts the inversion process. Next, we tested this mechanism using the MDS datasets. 

### 3.3. C. Physical Transformation during Ripple Inversion

To study the shape transformations, we first identified all the inversion events in a particular ripple simulation by plotting the central height over time, as shown in Figure 6a. Root-mean-square deviation (RMSD) analysis confirms the atom positions have stabilized within the first 0.2 ns. The initial time data are not included in the plot. After stabilizing, the ripple’s height is positive in the beginning, then flips 6 times during the next 8 ns. To track the larger structure of the ripple, we also recorded the height of the ripple along both the x-axis and y-axis every picosecond. From these datasets, we determined the (x, y) coordinate for the maximum height in time. Next, for the 6 ps prior to the first ripple curvature inversion, we plotted the (x, y) coordinate of the maximum height as shown in Figure 6b. The ripple maximum begins near the center of the ripple (0,0), then shifts to the edge of the ripple just before inversion. This process of starting near the center and moving toward the edge was repeated for all the ripple curvature inversions, as shown in Figure 6c–g. Given that a large displacement of the maximum away from center and toward the edge occurs just prior to inversion for every event shown in Figure 6, we believe that this displacement triggers ripple curvature inversion.

Next, we analyzed the shape of the ripple in time. This was carried out by using regression analysis on the cross-sectional datasets taken in both the x- and y-directions. For convenience, we redrew Figure 6a in Figure 7a. Cross-sectional datasets were extracted every picosecond and fit with both a cosine function with its maximum z-displacement at (0,0) and a sine function with its minimum z-displacement at (0,0). The resulting goodness-of-fit values (R^2^) are shown for each function in time, as the ripple inverts. Early on, the goodness-of-fit for cosine is close to one, while sine is close to zero, as shown in Figure 7b. Here, the ripple has the normal cosine shape. As time progresses, the quality of the cosine fit decreases, while the quality of the sine fit increases. Therefore, the ripple now takes on a shape more like that shown in Figure 5b. At the end of the time series, the cosine fit returns to nearly one, and the ripple has fully inverted. This sequence of shape changes, from cosine to sine and back to cosine, is repeated for all ripple inversion processes, as shown in Figure 7c–g. This change in the goodness-of-fit occurs in either the x- or y-direction; in some cases, it occurs in both directions. Only 8 ns of tracking data are shown in Figure 7a for conciseness. However, we completed the same analysis on 40 ns of data and confirmed the same correlation between the cross-sectional shape change and inversion. From these data, we believe that the shape transformation from cosine to sine, then back to cosine, is a key process for ripple curvature inversion.

## 4. Discussion

The primary path for the ripple curvature inversion process as determined by MDS is illustrated in Figure 8. We begin with a cosine-shaped convex ripple, as shown in Figure 8a. Due to random collective motion, the maximum of the ripple (marked with a red circle) begins to move away from the center and toward the fixed edge, as shown in Figure 8b. As the maximum moves to the right, the carbon atoms on the left side are pulled inward and downward toward the x–y plane (z = 0). These carbon atoms eventually cross the plane of the fixed outer edge. Now, the cross-sectional shape of the ripple is best described as a sine function (ripple minimum is marked with a green circle), as shown in Figure 8c. Finally, the ripple maximum suddenly flips from above the central fixed plane to below, and the ripple is concave, as shown in Figure 8d. This is the dominant pathway for ripple inversion. 

For comparison, we calculated the energy barrier from elastic theory [46,47,48]. To do this, we started with the ideal cosine shape and transitioned to the ideal sine shape shown in Figure 5b. The energy barrier from elasticity theory is significantly lower than what we found when the ripple simply flattens during the inversion process. However, elasticity theory overestimates the energy barrier by a factor of three to four compared to the results for MDS shown in Figure 4b. The reason for this is thought to be due to the sine shape being too ideal for what happens to the shape of the ripple as it passes through the fixed outer frame. Some evidence for this can be seen in the goodness of fit for the sine shape as it never reaches unity. One benefit of performing the MDS is that it removes the need to guess the precise shape of the ripple as it inverts its curvature.

The timescale for the curvature inversion process is remarkably short, at about 20–30 ps. During this time, the ripple height changes by 1.6 nm, which is significantly larger than the bond length. This yields an average speed of about 700 m/s. It is also noteworthy that the carbon atoms are moving together. This type of collective movement is like snap-through buckling [21,49,50,51]. This relatively large-area, high-speed movement has potential for small-scale energy harvesting [52,53].

We can utilize the MDS data to predict the current produced by the thermal movement of the graphene. We imagine forming a parallel plate capacitor with a time-varying capacitance given by $C\left(t\right)=\frac{\varepsilon A}{z\left(t\right)+1.5}$, where ε is the permittivity, $A=177\;{\mathrm{nm}}^{2}$ is the area of the graphene, and z(t) is the height of the graphene shown in Figure 7a; the fixed frame of the graphene is placed 1.5 nm away from the upper electrode shown in Figure 1. As the graphene moves, the distance between the ripple and the electrode varies with time, as shown in Figure 9a. This distance was chosen so that the graphene does not contact the electrode. The capacitance varies with time, as shown in Figure 9b. If we set the bias voltage to 1 V, then the charge, Q(t)=C(t) V, on the capacitor will also vary in time, as shown in Figure 9c. The rectified current varies with time, as shown in Figure 9d. Here, the current flowing is on the order of a nanoamp.

## 5. Conclusions

We studied the dynamics of graphene ripples using molecular dynamics simulations (MDS). The ripples are made by compressing flat graphene, then allowing the graphene to relax into the third dimension. For a range of strain values, the ripples are bistable and invert their curvature on a timeframe accessible with MDS. The transition rate for barrier crossing was found to be exponential in strain. Using the average transition rate, along with Kramers’ formula, we determined the height of the barrier in the double-well potential as a function of strain. The energy barrier is much lower than what one would expect if the ripples uniformly flattened before crossing. Using cosine and sine regression analysis on cross-sections of the ripple, we showed that the ripple distorts its shape to allow only part of the ripple to pass through the fixed frame at a time. Finally, we used the movement of the graphene to create a variable capacitor, which then induced a current to flow in a nearby circuit, thereby quantifying the energy harvesting promise of this system.

## Figures and Tables

**Figure 1 membranes-11-00516-f001:**
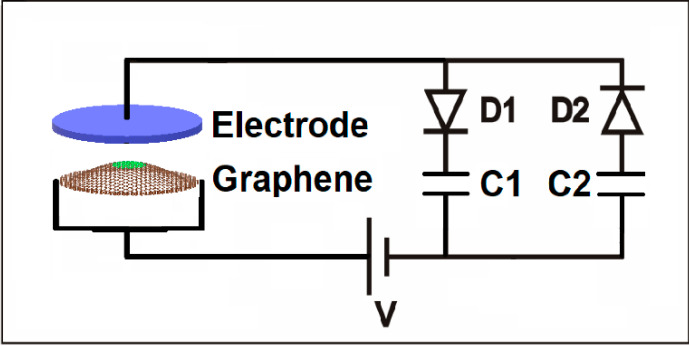
Graphene energy harvesting circuit using two diodes (D1 and D2), two storage capacitors (C1 and C2), and a rechargeable battery (V) to charge the graphene-electrode variable capacitor on the left.

**Figure 2 membranes-11-00516-f002:**
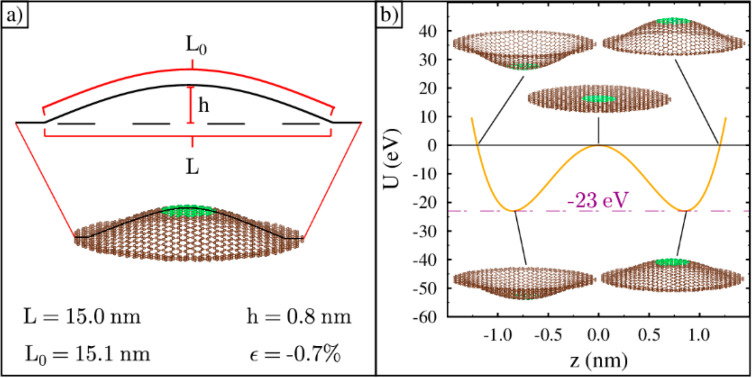
Ripple construction and the double-well potential: (**a**) An example graphene ripple constructed after compressing a flat sheet by 0.7% and adding a third dimension so that the arc length equals its original uncompressed length; (**b**) The double-well potential for the ripple. Five ripple illustrations show the shape at different points along the energy curve. The function precisely fits the five energy data points found at zero Kelvin by stretching and compressing the lattice by 0.7%.

**Figure 3 membranes-11-00516-f003:**
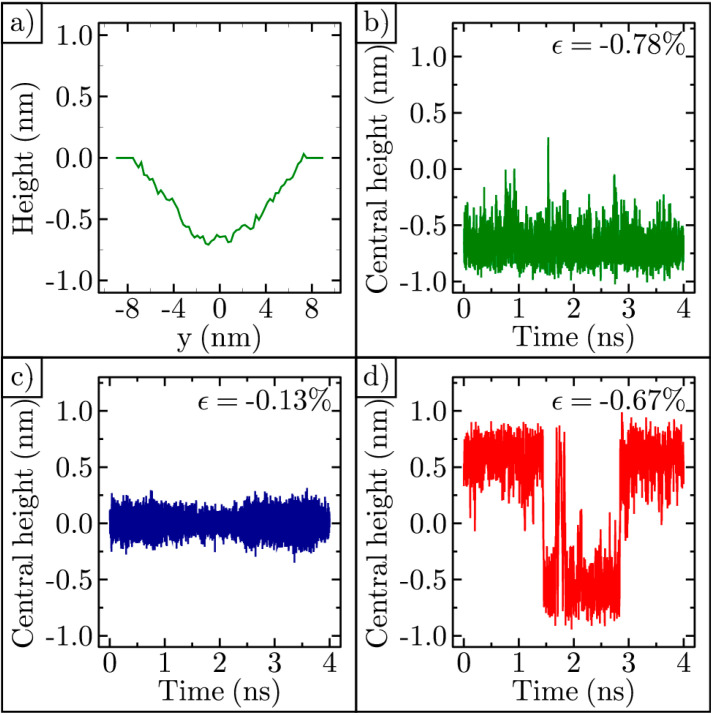
Role of strain on ripple dynamics at 3000 Kelvin: (**a**) Typical cosine-like cross-sectional profile of a ripple; (**b**) Central height of a highly compressed ripple in time; (**c**) Central height of a lightly compressed ripple in time; (**d**) Central height of a bistable ripple in time.

**Figure 4 membranes-11-00516-f004:**
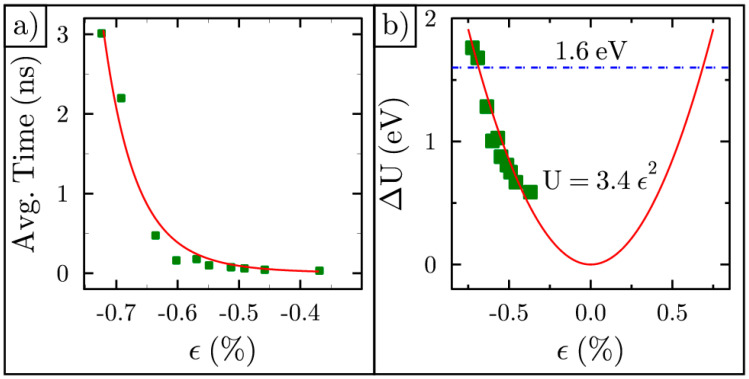
Average time and energy barrier: (**a**) Average time between curvature inversion vs. strain; (**b**) Barrier height vs. strain.

**Figure 5 membranes-11-00516-f005:**
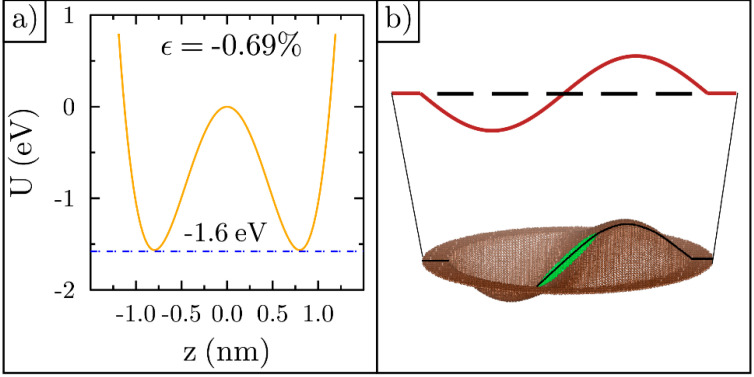
Potential energy and ripple shape: (**a**) Double-well potential for strain of −0.7%; (**b**) Illustration of a possible ripple shape during inversion, along with its cross-section.

**Figure 6 membranes-11-00516-f006:**
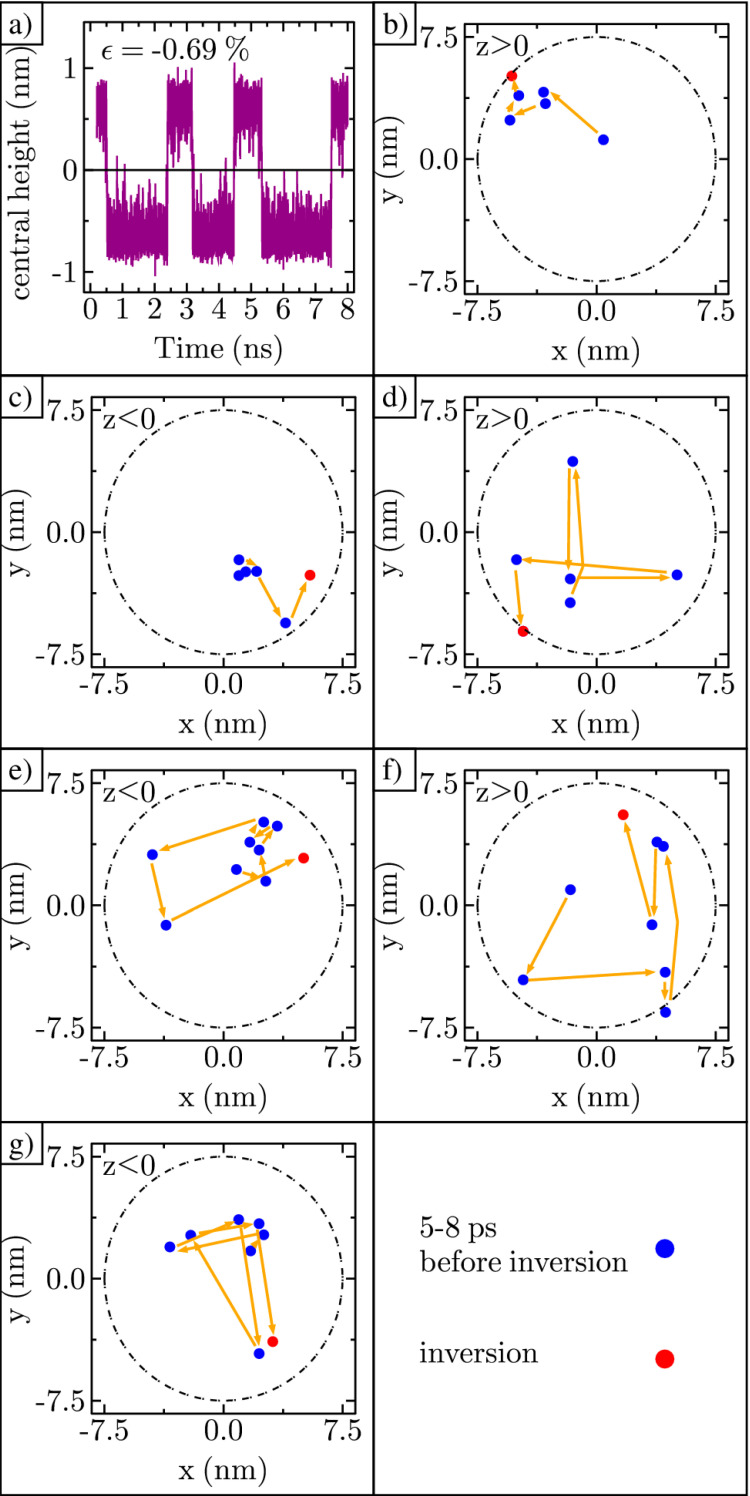
Ripple maximum shifts toward edge: (**a**) Central height of ripple in time with six complete inversions; (**b**–**g**) Displacement of the maximum along the x and y axes every picosecond, just before curvature inversion. Lower right box shows the legend.

**Figure 7 membranes-11-00516-f007:**
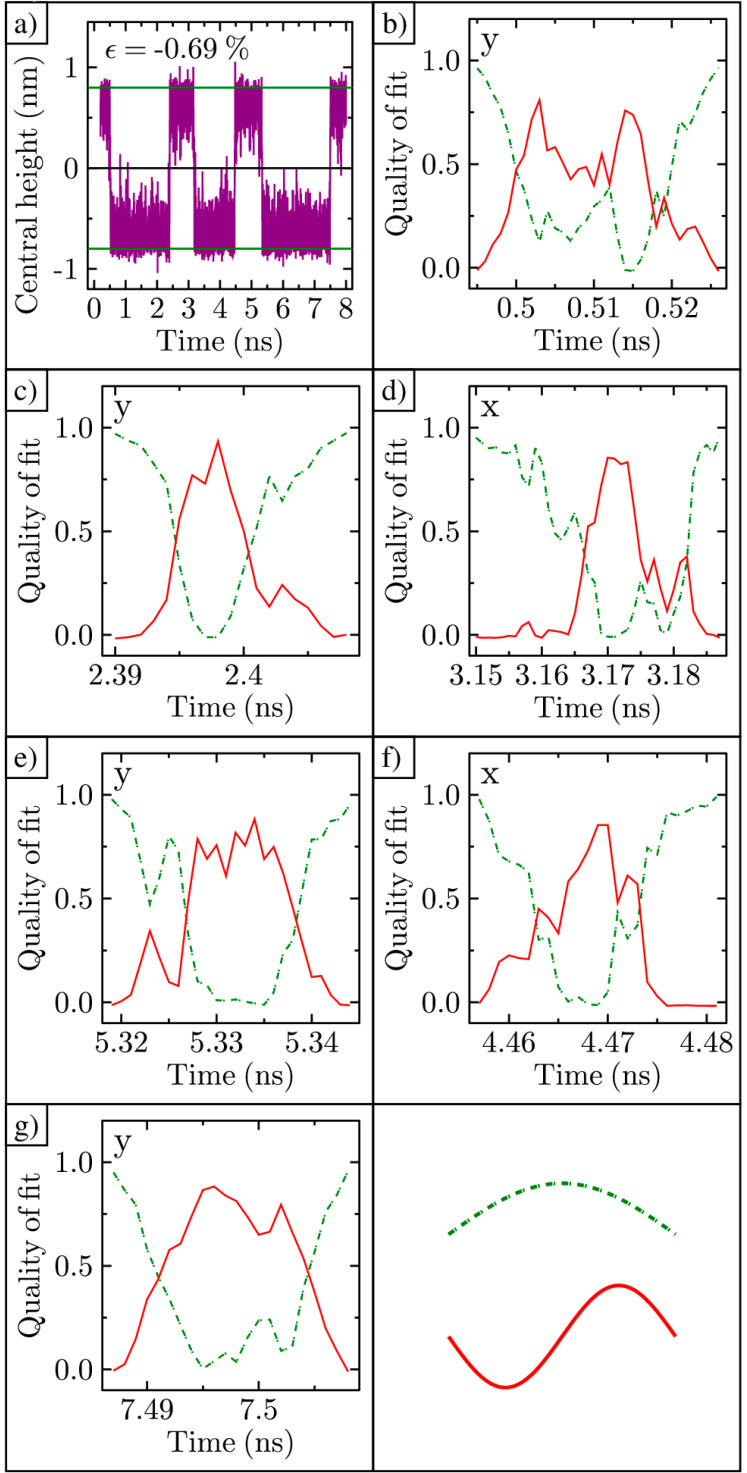
Cosine–sine–cosine shape transformation: (**a**) Central height of ripple in time; (**b**–**g**) Cosine and sine goodness-of-fit R^2^ values for the x- or y-direction (as labeled) cross-section of the ripple in time. Lower right box shows the legend.

**Figure 8 membranes-11-00516-f008:**
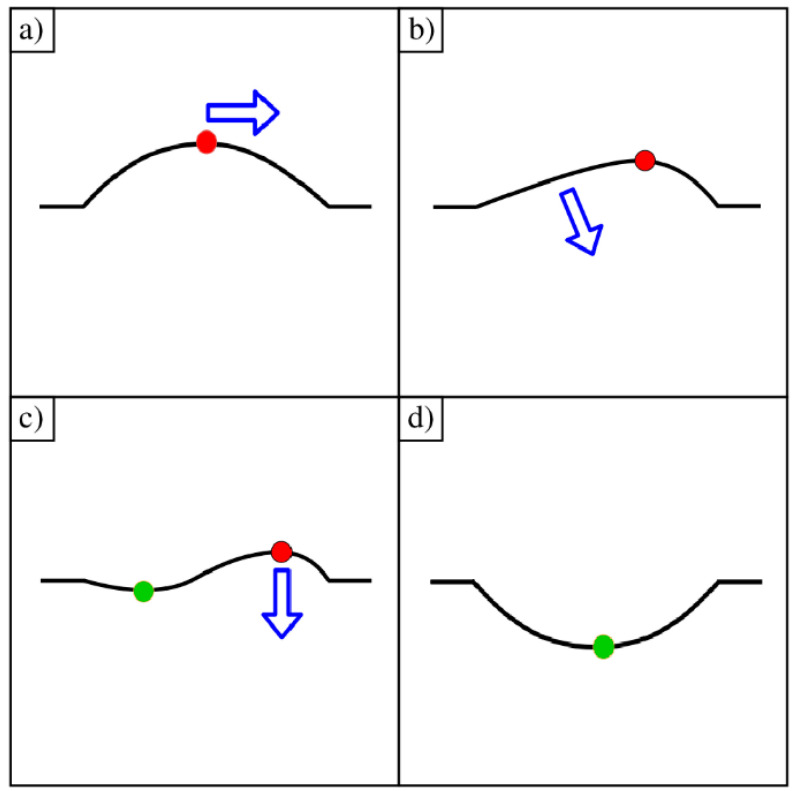
Conceptual illustration of the shape transformation through curvature inversion: (**a**) The maximum of the ripple moves from center to right; (**b**) Curvature is low on the left side. Local atoms are pulled down toward the fixed frame; (**c**) The left side passes through the x–y plane and the ripple forms a sine-like shape; (**d**) The left sub-ripple has pulled the right sub-ripple down. A single ripple is formed below the x-y plane.

**Figure 9 membranes-11-00516-f009:**
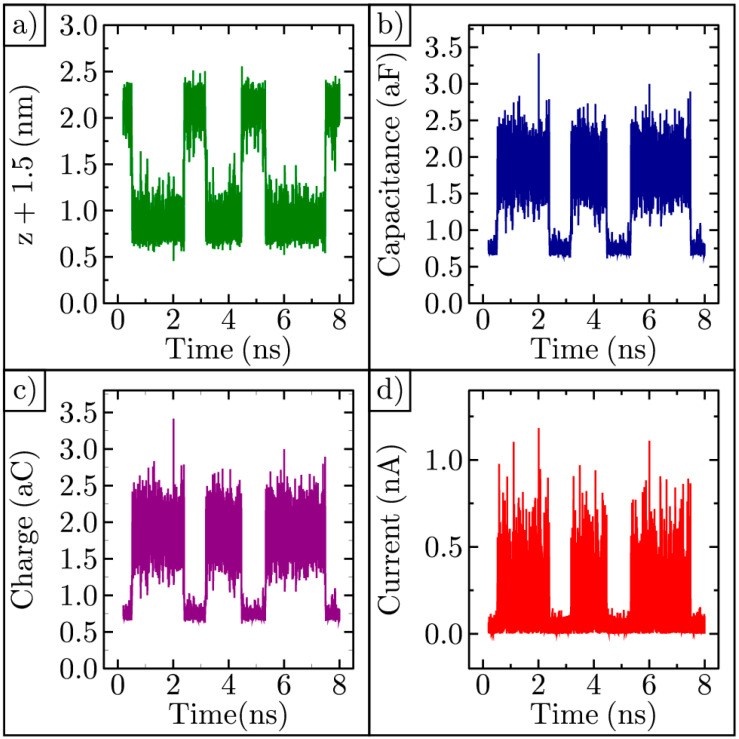
Graphene ripple near a fixed electrode: (**a**) The distance between the electrode and the ripple; (**b**) Capacitance of the graphene-electrode capacitor; (**c**) Charge on the graphene capacitor; (**d**) Rectified portion of the alternating current flowing in the circuit.

## Data Availability

The data that support the findings of this study are available from the corresponding author upon reasonable request.

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
