# Peer review of "Mechanisms of Spontaneous Curvature Inversion in Compressed Graphene Ripples for Energy Harvesting Applications via Molecular Dynamics Simulations"

_membranes, 2021, doi:10.3390/membranes11070516_

Round 1
Reviewer 1 Report
The paper entitled "Mechanisms of spontaneous curvature inversion in compressed graphene ripples for energy harvesting applications via molecular dynamics simulations" contains interesting work using molecular dynamics to investigate the use of graphene ripples for in energy harvesting applications. The work is creatively designed and done with the right tools. But it needs a major revision. The following points can improve the quality of work: 1) Innovations should be highlighted in the introduction 2) Simulation time is short. To make sure the results are time-independent, the simulation needs to be repeated over and over again in the larger time scales. 3) Use RMSD analysis to show the stability of the simulations 4) To validate the data, try to simulate a similar reference work using your method (potential, force field and software) you used in this work and compare your results with the reference data. 5) Write more about simulation and optimization details 6) The quality of the figures is low and should be improved 7) The language of the text can be improved 8) Discuss more about the electronic and magnetic properties of graphene membranes. Especially in the introduction. 9) It is better to use DFT calculations to prepare additional analyzes to check the electronic properties of this work.
Author Response
Response to Reviewer 1 Comments
Point 1: Innovations should be highlighted in the introduction.
Response 1: We agree with the reviewer and have highlighted the innovations in the introduction. Please see additional text on lines 41-47.
Point 2: Simulation time is short. To make sure the results are time-independent, the simulation needs to be repeated over and over again in the larger time scales.
Response 2: We agree with the reviewer, and we have extended the analysis from 8 ns to 40 ns for each ripple in the study. Analysis of the larger time scales yields the same results reported for the earlier time scales. Please see additional text on lines 230-232.
Point 3: Use RMSD analysis to show the stability of the simulations.
Response 3: We agree with the reviewer and have carried out RMSD analysis. We confirmed that the simulations stabilize within the first 0.2 ns. Please see additional text on lines 200-202.
Point 4: To validate the data, try to simulate a similar reference work using your method (potential, force field and software) you used in this work and compare your results with the reference data.
Response 4: We agree with the reviewer. We have successfully reproduced the graphene ripple curvature inversion simulation results published here PRL 117, 126801 (2016). Please see additional text on lines 163-165.
Point 5: Write more about simulation and optimization details.
Response 5: We agree with the reviewer. Please see additional text on lines 95-96 and 106.
Point 6: The quality of the figures is low and should be improved.
Response 6: We agree with the reviewer. High resolution images have been uploaded to the journal. No change to text.
Point 7: The language of the text can be improved.
Response 7: We agree with the reviewer. Please see modifications to the text on lines 154, 174, 186, 198, 202, 229, 262, and 269-270.
Point 8: Discuss more about the electronic and magnetic properties of graphene membranes. Especially in the introduction.
Response 8: We agree with the reviewer. Please see additional text on lines 41-47.
Point 9: It is better to use DFT calculations to prepare additional analyzes to check the electronic properties of this work.
Response 9: We agree with the reviewer. DFT calculations for graphene membranes were carried out in another study published here PRB 84, 161401(R) (2011). Please see additional text on lines 134-135 and on lines 187-188.
Reviewer 2 Report
The authors simulated the molecular dynamics of compressed graphene membranes thoroughly. The authors found the range of compressive strain in which curvature inversion occurs, and they revealed the mechanism for curvature inversion using different means of data analysis. A potential application of the studied graphene membranes is also discussed. The paper is well-written and convincing. I am glad to recommend its publication in Membranes.
One minor comment: the authors might consider changing “-1.6 eV” to “1.6 eV” in line 177.
Author Response
Response to Reviewer 2 Comments
Point 1: Change “-1.6 eV” to “1.6 eV” in line 177.
Response 1: We agree with the reviewer and have made this change.
Reviewer 3 Report
Comments: The author built numerous three-dimensional graphene ripples, with each featuring a different amount of compression, and performed moleculardynamics simulations at elevated temperatures. Finally, the author used the movement of the graphene to create a variable capacitor, which then induced a current to flow in a nearby circuit, thereby quantifying the energy harvesting promise of this system. This work is very innovative and interesting. I highly recommend it to be published as it is.
Author Response
No changes required.
Round 2
Reviewer 1 Report
The authors have addressed my comments, I thus recommend the acceptance of this manuscript for the publication